# Multi-Agent Multi-View Collaborative Perception Based on Semi-Supervised Online Evolutive Learning

**DOI:** 10.3390/s22186893

**Published:** 2022-09-13

**Authors:** Di Li, Liang Song

**Affiliations:** 1College of Information Engineering, Henan University of Science and Technology, Luoyang 471000, China; 2Academy for Engineering & Technology, Fudan University, Shanghai 200433, China

**Keywords:** semi-supervised learning, online evolutive learning, collaborative perception, discriminative information fusion

## Abstract

In the edge intelligence environment, multiple sensing devices perceive and recognize the current scene in real time to provide specific user services. However, the generalizability of the fixed recognition model will gradually weaken due to the time-varying perception scene. To ensure the stability of the perception and recognition service, each edge model/agent needs to continuously learn from the new perception data unassisted to adapt to the perception environment changes and jointly build the online evolutive learning (OEL) system. The generalization degradation problem can be addressed by deploying the semi-supervised learning (SSL) method on multi-view agents and continuously tuning each discriminative model by collaborative perception. This paper proposes a multi-view agent’s collaborative perception (MACP) semi-supervised online evolutive learning method. First, each view model will be initialized based on self-supervised learning methods, and each initialized model can learn differentiated feature-extraction patterns with certain discriminative independence. Then, through the discriminative information fusion of multi-view model predictions on the unlabeled perceptual data, reliable pseudo-labels are obtained for the consistency regularization process of SSL. Moreover, we introduce additional critical parameter constraints to continuously improve the discriminative independence of each view model during training. We compare our method with multiple representative multi-model and single-model SSL methods on various benchmarks. Experimental results show the superiority of the MACP in terms of convergence efficiency and performance. Meanwhile, we construct an ideal multi-view experiment to demonstrate the application potential of MACP in practical perception scenarios.

## 1. Introduction

In the edge intelligence [1,2,3] environment, many sensing devices recognize the local scene to provide corresponding smart services in real-time. However, most current intelligent sensing applications still rely on a fixed recognition model or a unified cloud model [4]. Since the perception scene changes over time, the feature distribution of the perception data will continue changing, and the generalizability of the fixed recognition model will be highly affected. The degradation of model generalizability will significantly impact the service quality of edge agents/models. Relying on regular manual annotations for model tuning will incur high ongoing costs and increase the deployment difficulty.

To reduce manual annotation and continuously improve the adaptability of each edge model to perception data changing, it is required that each edge model/agent is effectively able to use the newly added unlabeled perceptual data to conduct online training and model tuning unassisted [5], forming an online evolutive learning [6,7] (OEL) system. The semi-supervised learning [8,9] (SSL) method can utilize the knowledge learned from a small amount of labeled data and dig the adequate discriminative information from massive unlabeled amounts of data to achieve continuous model optimization, which effectively fits the online training strategy. At the same time, due to the characteristics of multi-view edge sensing devices, multiple sensing models can obtain perception data from different viewpoints with the same semantic target, and these perception data have pieces of strong complementary information. Through the collaborative discrimination and information fusion of multi-model perception data from different views, the discriminative reliability of edge models for unlabeled data will be greatly improved, and the confirmation bias [10] problem in the SSL process will be significantly reduced. Such an SSL-based OEL method with multi-model collaborative perception and continuous self-training will enable each edge sensing model to enhance its adaptability to data distribution changes, continuously improve model generalization without relying on manual annotations, and reduce the deployment complexity of the perception tasks.

SSL has developed rapidly [11,12] in recent years and has gradually become the primary method for solving label-scarcity problems and adapting data distribution for practical applications. However, multi-model and multi-view SSL methods [13] are still relatively rare. The existing multi-view SSL methods [14] are generally only for fixed small multi-view datasets, and the multi-view data are typically obtained through different feature extractors. Such multi-view data present challenges to guaranteeing the view independence between features, which makes these methods unable to be generalized to practical recognition tasks. In terms of multi-model SSL, the method of constructing multiple views from single-view data is relatively simple, and there are rare methods for multi-view agents to perceive scenes and solve practical OEL tasks.

Maintaining the discriminative independence between multi-view models and improving the model’s collaborative discrimination reliability is the key to semi-supervised online evolutive learning systems. In this paper, we propose a multi-view agent’s cooperative perception (MACP) semi-supervised online evolutive learning method that can solve the OEL problem well in the multi-view perception environment. Specifically, we first use different self-supervised [15] model-initialization (SMI) methods for different edge models, so that they can learn differentiated feature-extraction patterns from various self-supervised tasks. Combined with different data-normalization methods, SMI ensures the discriminative independence of each model when facing view-specific data. We then propose a discriminative information fusion (DIF) algorithm that votes and integrates the multi-view model’s predicted distributions. DIF obtains a more reliable discriminant representation from unlabeled data for model training based on the discriminant criteria differences between multi-view models. To maintain the differentiation of the discriminant standards during the training process of each model, we further propose a parameter constraint (PC) method between models. By orthogonalizing some critical parameters of different models, the discriminant standards of each model are effectively prevented from converging during the training process. The proposed MACP achieves better convergence efficiency and final performance than other representative single-model SSL and multi-model SSL methods on multiple datasets. At the same time, we find that the pseudo-labels obtained by multi-view models based on DIF maintain a high accuracy rate during the training process, which further proves the reliability of our method. Moreover, since our method mainly constructs multi-view data from a single-view dataset, to explore the performance of MACP under an ideal multi-view sensing environment, we configure a collaborative perception experiment where the perception data streams are different data with the same categories. In an ideal multi-view perception environment, MACP achieves a performance that surpasses fully supervised learning methods under the same configuration, demonstrating our method’s application potential in practical multi-agent collaborative sensing.

The main contributions of this paper can be summarized as:We analyze the existing problems of multi-agent collaboration and data-distribution adaptation in the multi-view sensing environment. We propose the multi-view agent’s collaborative perception (MACP) semi-supervised online evolutive learning method. MACP can reduce the task complexity of multi-model SSL when processing multi-view perception data and realize real-time tuning of the local perception system.In MACP, we enable each model to learn a differentiated feature-extraction mode through a self-supervised model-initialization method, which enhances the discriminative independence of each model. By applying the discriminative information-fusion approach to the predictions of each view model, the reliability of the discriminant results is improved, and continuous consistency regularization training is realized. Through further regularization constraints on the parameters of each model in the training process, the model can continue to maintain a relatively independent discriminative ability, and the stability of the entire OEL system is improved.The proposed MACP achieves a better performance than the comparison methods on multiple datasets. In an ideal multi-view agent collaborative perception experiment, MACP exceeds the performance of the fully supervised learning method, which proves the applicability of the proposed method in practical multi-view sensing scenarios.

## 2. Related Work

Consistency-based [16,17], semi-supervised learning methods have achieved great success in recent years. The main theoretical basis is that the model should maintain consistent predictions for different input variants with the same semantics. Based on such a natural constraint, using data-augmentation methods to transform the input features and train the model to mine the consistency information of different input variants can effectively improve the generalization of the SSL model. MixMatch [18] first performs multiple augmentation operations on the same unlabeled input, then uses sharpening [19] on the average of all augmented data predictions, and finally guides the SSL model training through the prediction targets generated by mixing up [20] data and labels. FixMatch [21] simplifies the complex process of the previous work and directly inputs weakly augmented and strongly augmented versions of the same unlabeled data into model training. Better performance is achieved by converting the higher confidence part of the weakly augmented predictions into pseudo-labels to guide the model training on the strongly augmented data. AWLDA [22] proposes a strategy to count the class-wise learning progress in the training process and improves the contribution of hard-to-learn classes to training, which reduces the class imbalance problem in the SSL process. Meanwhile, this method makes better use of the consistent relationship between low-confidence predictions and significantly improves the convergence speed of SSL. Since this paper mainly studies the application of multi-view and multi-model semi-supervised learning methods in OEL scenarios, we will focus on analyzing research related to these goals.

For multi-view, semi-supervised learning methods, Co-training [23] first trains two different classifiers with labeled data of different views, and then exchanges the high-confidence predicted results for unlabeled data between each classifier for SSL training, realizing the discriminative information-sharing of each view model. DCT [24] proposes a differential constraint method based on adversarial samples for the co-training models, which makes the discriminative criteria of each model’s approach to the adversarial samples of each other while providing mutual annotations. This method can continuously improve the discriminative difference of different view models but adds certain extra computation. Tri-training [25] first proposes a multi-view training method that does not rely on view differences. It uses bootstrap to sample three different subsets of data from the labeled dataset to train three different initial classifiers and then performs a voting process on the prediction results of the unlabeled data. The predictions agreed by the majority classifier will be used as the training target of the minority classifier. Tri-net [26] introduces the output-smearing [27] process for the three models with shared parameters to maximize the prediction difference of each model and then uses the voting results of the two models’ predictions as the training target for the training of another model. This approach requires the periodic fine-tuning of the discriminative variability of the models during training, thus potentially reducing the coherence of implementation. Ref. [28] proposes a multi-view, semi-supervised feature-representation learning method that utilizes orthogonalization and adversarial constraints to improve the consistency between models and the ability to extract complementary information. Other graph-based multi-view SSL methods [29,30] achieve better results on various specific multi-view databases by learning the relationship between feature representations from different views.

For multi-model, semi-supervised learning, such methods mainly utilize knowledge transfer between models in different learning states to achieve common performance improvements. The Π-model [31] achieves steady performance gains by using two models to predict different variants of the same unlabeled data and attaching consistency constraints to the predicted distributions. Meanteacher [32] adopts the mechanism of a teacher–student model for comparative learning and uses the exponential moving average of the updated parameters of the student model in the past training process as the teacher model. During training, the similarity between the discriminative results of the teacher model and the student model on unlabeled data is continuously strengthened. Dualstudent [33] believes that the teacher model in the aforementioned work is the historical average of the parameters of the student model, so there may be a performance bottleneck when guiding the training of the student model. It proposes a dual-student model structure, which further improves the model performance by evaluating the uncertainty of the prediction results of the two models and adding mutual stability constraints to the high-reliability predictions. MPL [34] proposed a teacher–student model structure based on the idea of meta-learning. The prediction of the teacher model on unlabeled data is used as the training target of the student model, and the classification loss of the student model on the labeled data is used as the training feedback for the teacher model. Such an information-interaction method enables the teacher model to optimize the pseudo-label discrimination criteria continuously.

The main work of existing multi-view or multi-model SSL algorithms [35] is to improve the discriminative difference between models and then mine complementary information between models to enhance the reliability of unlabeled data prediction. However, these methods generally have problems, such as the complicated design of differential constraint strategy and the insufficient universality of the method. For multi-model collaboration in OEL scenarios, more comprehensive research is needed.

## 3. Proposed Methods

In this section, we first provide the problem definition and state the perception environment and main goals of a multi-view agent learning system. Then, the realization method of each part-module of MACP is introduced. We represent view-specific agents with multiple different SSL models and utilize a continuous unlabeled data stream to simulate a multi-agent sensing environment.

### 3.1. Problem Definition

As shown in Figure 1, in a multi-agent perception environment, we can let $M_{1},M_{2},\ldots ,M_{v}$ be a set of multi-view edge models, where *v* represents different views. The models from different viewpoints will perceive the same scene in real-time.

For a semi-supervised multi-view classification task, let $X^{v}=\left(x_{i}^{v},y_{i}\right),i\in \left(1,\ldots ,B\right)$ represent a batch of *B*-labeled training data $x_{i}^{v}$ of the SSL model with view *v*, where $y_{i}$ is the unified label of all view data. Let $U^{v}=u_{i}^{v},i\in \left(1,\ldots ,\mu B\right)$ represent a batch of μB unlabeled perception data $u_{i}^{v}$ of the SSL model with view *v*, where μ is a hyperparameter that controls the proportion of labeled and unlabeled data. Note that, due to the fixed view of the training dataset, the $X^{v}$ and $U^{v}$ of models with different views are generated by random image-augmentation methods $A_{w}\left(X\right)$ and $A_{w}\left(U\right)$, respectively, where $A_{w}\left(\cdot \right)$ represents applying a random weak augmentation transformation to the input. The total amount of labeled data $X^{v}$ for each view model will be much less than the unlabeled data $U^{v}$. In each training iteration, the models $M_{v}$ from different views will use real-time generated data streams $X^{v}$ and $U^{v}$ for SSL training, and the high-confidence pseudo-label of $U^{v}$ will be determined by the collaborative discrimination of each view model.

Let $P_{M_{v}}\left(y|x\right)$ represent the prediction result of the model $M_{v}$ for the input data *x*. The main goal of the multi-view agent’s collaborative perception (MACP) method is to fuse the discriminative results of each view model $M_{v}$ on the unlabeled perception data $U^{v}$ to obtain more reliable pseudo-labels for model training on their respective view data. As a result, the generalizability of each view model and its adaptability to the data distribution changes will continue to improve. Maintaining the discriminative independence of each view model during the training process will be the key to improving the reliability of the collaborative discriminant results.

### 3.2. Overall Framework

The main task of MACP is to design an effective collaborative discriminant process, which uses the multi-view model to predict perceptual data and achieve high-reliability pseudo-label extraction from the view-specific predictions. Each view model uses the collaborative discrimination results for continuous perception and training, so that the overall learning system can continuously adapt to changes in data distribution.

The overall framework of the proposed method is shown in Figure 2. Note that, in addition to a small amount of labeled data, a large amount of unlabeled perceptual data will continue to feed into different view models, constituting a continuous online learning process. MACP mainly includes three steps. First, the models from different views are initialized based on different self-supervised learning methods to ensure that each model has a differentiated feature-extraction pattern. Then, each model performs discriminative information-fusion processing on the predictions of view-specific perceptual data to obtain high-confidence pseudo-labels, which are used in the consistency-regularization training process of SSL. Finally, additional parameter constraints are introduced into the model-training process to maintain the discriminative independence of each model during the training, thereby preserving the stable operation of the entire learning system.

### 3.3. Self-Supervised Model Initialization

Since models from different views will predict different perceptual data with the same semantics, each model’s discriminative independence will significantly impact the final discriminative information-fusion results. The more independent discriminant ability will make each model make mistakes in different places, so the obtained fusion discriminant results will have higher reliability. We use various initialization methods for models from different views to obtain each model’s view-specific feature-extraction pattern in the model initialization stage.

Specifically, taking the three-view perception models set as an example, for the first-view model, we only perform default parameter initialization processing for it. For the models from the other two views, we pre-train them on the self-supervised learning-based jigsaw puzzle solving [36,37] task and the generative adversarial network [38] task, respectively, to increase the differences in the feature-extraction patterns among the models.

For the self-supervised jigsaw-puzzle-solving task, let $U^{v}$ be the self-supervised training data. For each $u_{i}^{v}$ in $U^{v}$, we slice it into *N* image patches of equal size and assign labels $y^{n}$ to each patch in order. Then, we randomly shuffle the image patches and stitch them into a new image ${\hat{u}}_{i}^{v}$. Through an *N*-way classifier, the model $M_{v}$ will predict the position of each image patch in the spliced image, and the label $y^{n}$ will be used to guide the model to generate the correct image patch order prediction for the scrambled image. The loss function of the jigsaw puzzle solving task is as follows:(1)$$L_{jigsaw}=-\frac{1}{\mu BN}\sum_{b=1}^{\mu B}\sum_{n=1}^{N}y^{n}log\left(P_{M_{v}}{\left(y|{\hat{u}}_{b}^{v}\right)}^{n}\right),$$
where $P_{M_{v}}{\left(y|{\hat{u}}_{b}^{v}\right)}^{n}$ represents the category prediction of model $M_{v}$ for the *n*-th image block in ${\hat{u}}_{b}^{v}$, and $L_{jigsaw}$ is the loss function of *N*-way categorical cross-entropy for all images in the current batch *B*. By solving the jigsaw puzzle task, the view-specific model $M_{v}$ can learn a good representation of the spatial positional relationship of the image, thereby focusing on extracting differentiated features that are different from other view models.

For the self-supervised generative adversarial network (GAN) task, while maintaining the basic GAN composed of the generator and the discriminator, we train the generator to continuously learn the ability to generate images from the current database. At the same time, we modify the discriminator so that it is not only responsible for predicting the authenticity of the generated images, but also has image-classification capabilities. Specifically, for the original classification model $M_{v}$, we keep its basic classifier unchanged and perform additional activation processing on the output logits $z_{c}$ of the model:(2)$$q_{binary}\left(z_{c}\right)=\frac{\sum_{c=0}^{C}exp\left(z_{c}\right)}{\sum_{c=0}^{C}exp\left(z_{c}\right)+1}\;,$$
where *C* is the number of categories of the original image classifier, and, through exponential normalization, the multi-dimensional output $z_{c}$ of the model for a certain input is converted into a one-dimensional binary prediction $q_{binary}\left(z_{c}\right)$, which is used for the training of the discriminator.

We know that in Equation (Equation 2), when the value of $z_{c}$ is relatively large, $q_{binary}\left(z_{c}\right)$ will be close to 1, and when the value of $z_{c}$ is relatively small, $q_{binary}\left(z_{c}\right)$ will be close to 0. This additional activation can train the discriminator to predict larger logits for real images and smaller logits for generated fake images, enabling the co-training of the discriminator for both multi-classification and generative adversarial tasks. The loss function of the self-supervised GAN task is as follows:(3)$$L_{gan}=-\frac{1}{\left|X\right|}\sum_{x\in X}ylog\left(P_{M_{v}}\left(y|x\right)\right)+\frac{1}{\left|D\right|}\sum_{x\in D}BCE\left(y_{b},q_{binary}\left(P_{M_{v}}\left(y|x\right)\right)\right).$$

In Equation (Equation 3), the first term is the categorical cross-entropy loss for the labeled dataset X, and the second term is the discriminator’s binary cross-entropy loss for the real images and generated images in the entire dataset *D*. After the training of the self-supervised GAN, the model $M_{v}$ will focus on mining the essential feature representation related to image generation, so as to gain a differentiated feature-extraction ability.

Through the designed self-supervised learning task, the view-specific initialization of each model is realized, and models from different views will use different feature-extraction patterns to predict the perceptual data. This paper takes the three-view model as an example to illustrate the self-supervised model-initialization (SMI) process. The SMI of more views can be implemented by using increasingly different self-supervised tasks such as image colorization [39], image super-resolution [40], contrastive learning [41], etc.

### 3.4. Discriminative Information-Fusion

The discriminative results of relatively independent multi-view perceptual data contain the consensus and complementary information of the predicted target. The effective fusion of these predicted distributions can obtain a more accurate class representation for the consistency regularization of SSL training.

The discriminative information-fusion process of the multi-view agent is shown in Figure 3. First, for each view *v*, we perform high-confidence filtering on the predictions of model $M_{v}$ on the current batch of perceptual data $U^{v}=u_{i}^{v},i\in \left(1,\ldots ,\mu B\right)$ to obtain predicted class labels and their corresponding indices for samples that satisfy the threshold condition:(4)$${\left[I^{v},C^{v}\right]}_{v\in V}=\sum_{i=1}^{\mu B}1\left(max\left(P_{M_{v}}\left(y|u_{i}^{v}\right)\right)>\tau \right)\cdot \left(argmax\left(P_{M_{v}}\left(y|u_{i}^{v}\right)\right)\right),$$
where $1(max\left(P_{M_{v}}\left(y|u_{i}^{v}\right)\right)>\tau )$ represents fetching the predicted class distributions greater than the threshold τ from the predictions of $U^{v}$, and $argmax(P_{M_{v}}\left(y|u_{i}^{v}\right))$ means obtaining the category label with the maximum probability in the corresponding prediction result. The predicted class labels and indices of high confidence predictions in the current batch are obtained through the above processing. $I^{v}$ and $C^{v}$ are two vectors that store the indices and class labels of valid samples obtained from the perceptual data of the current view *v*, and *V* is the total number of views.

Using Equation (Equation 4), we extract the indices of high-confidence samples and their corresponding class labels ${\left[I^{v},C^{v}\right]}_{v\in V}$ in the prediction results of each view model. These results will be used for voting and aggregating to perform discriminative information fusion. The voting and aggregating process can be expressed as:(5)$${\left[{\tilde{I}}^{v},{\tilde{C}}^{v}\right]}_{v\in V}=\left[I^{v},C^{v}\right]\cap \left[I^{\neg v},C^{\neg v}\right],v\in \left(1,\ldots ,V\right),$$
(6)$$\left[I,C\right]=\left[{\tilde{I}}^{v},{\tilde{C}}^{v}\right]\cup \left[{\tilde{I}}^{\neg v},{\tilde{C}}^{\neg v}\right],v\in V.$$

In Equation (Equation 5), for the index- and class-label vectors $\left[I^{v},C^{v}\right]$ of each view model, we, respectively, intersect them with the results of other view models to obtain the prediction consensus. Where $\left[I^{\neg v},C^{\neg v}\right]$ represents the sample indices and class labels of other view models, and ${\left[{\tilde{I}}^{v},{\tilde{C}}^{v}\right]}_{v\in V}$ are each view model’s voted results. Then, according to Equation (Equation 6), we take the union of all the compatible parts of ${\left[{\tilde{I}}^{v},{\tilde{C}}^{v}\right]}_{v\in V}$ to obtain the final discriminative fusion result [I,C].

Then, the discriminative fusion results will be used by each view model for SSL training. For the unlabeled data $U^{v}$ of each view, we first obtain the corresponding samples according to the index *I* and perform strong data-augmentation processing on them:(7)$$\tilde{U^{v}}=A_{s}\left(u_{i\in I}^{v}\right),$$
where $A_{s}\left(\cdot \right)$ represents the random strong data-augmentation function, and $u_{i\in I}^{v}$ represents the samples extracted from $U^{v}$ according to the index *I*. At this time, the unlabeled perception data of each view in the current batch that meet the conditions will be assigned a pseudo-label $C_{i}$ from *C*, and the unlabeled data batch becomes $\tilde{U^{v}}=\left({\tilde{u}}_{i}^{v},C_{i}\right),i\in \left(1,\ldots ,|C|\right)$.

Based on the discriminative information fusion (DIF) of the multi-view models, the final SSL loss of each view model can be expressed as:(8)$$L_{ssl}^{v}=\frac{1}{B}\sum_{i=1}^{B}H\left(y_{i},P_{M_{v}}\left(y|x_{i}^{v}\right)\right)+\frac{1}{\left|C\right|}\sum_{i=1}^{\left|C\right|}H\left(C_{i},P_{M_{v}}\left(y|{\tilde{u}}_{i}^{v}\right)\right),$$
where H(y,x) represents the categorical cross-entropy loss, the first term of Equation (Equation 8) is the supervised loss of the labeled data $x_{i}^{v}$ under the current batch, and the second term is the unsupervised loss for the augmented unlabeled data ${\tilde{u}}_{i}^{v}$ with the pseudo-label $C_{i}$ as the target. In each iteration of the different view models, reliable pseudo-labels are obtained by collaborative DIF of perceptual data for their respective unsupervised loss calculations. Multi-view agents can fully use their different discriminant criteria to better mine generalization information from multi-view perception data.

### 3.5. Parameter Constraint

Since models from different views will use the pseudo-labels provided by DIF to train on their own perception data, as the model reaches higher iterations, the discrimination independence provided by SMI will gradually weaken. Therefore, the discriminative criteria of each view model may have the convergence risk in the later training stages. To prevent the increase of confirmation bias during model training, we further introduce a parameter-regularization constraint for different view models, making each model maintain discriminative independence as much as possible during the training process.

Specifically, we sequentially perform orthogonalization constraints on the critical parameters of each view model. During the training process, the parameters of the output layer and the critical feature-extraction layer of each model are kept irrelevant, thereby reducing the possibility of model-discriminating pattern convergence.

Let the critical parameters of the model $M_{v}$ be $\Theta_{v}$. The additional parameter-constraint loss can be expressed as:(9)$$L_{reg}^{v}={\left|\sum \left(\Theta_{v}\cdot \Theta_{v-1}\right)\right|}_{v\neq 1},$$
where $\Theta_{v-1}$ is the critical parameter of the previous view model, and the inner product loss of the two sets of critical parameter vectors will ensure that the parameters of different view models are orthogonalized. Through parameter constraints, the model of each view will gradually increase the differentiated discriminative ability during the training process.

Combining the SSL loss and parameter constraint loss, the total loss function of each view model in MACP is:(10)$$L^{v}=L_{ssl}^{v}+L_{reg}^{v}.$$

## 4. Experiments

This section will first introduce the implementation details and hyperparameter configuration of the proposed method and report the performance comparison and efficiency analysis of MACP with other representative multi-model and single-model SSL algorithms. Then, we configure an experiment in an ideal multi-view perception environment to illustrate the effectiveness of MACP in practical OEL applications. In the ablation study, the effects of different modules of MACP on the training performance are analyzed, and the variants in discriminative information-fusion methods are discussed.

### 4.1. Implementation Details

Since several representative SSL algorithms are configured in different experimental environments, to ensure a fair comparison, we re-implement several methods used for performance analysis in the same environmental configuration, while ensuring that the training hyperparameters are as similar as possible. Our main programming environment is the Keras deep learning library with Tensorflow as the backend.

**Experiment Settings** We configure two experiments to demonstrate the effectiveness of the proposed method. When comparing with general multi-model or single-model SSL methods, we train the model multiple times with different labeled-data splits and compare the test-error rate with other methods. Since the existing multi-view datasets are generally small and unrepresentative, in order to better reflect the performance of the proposed method in a realistic perception environment, we then use the existing dataset to simulate an ideal multi-view perception experiment. Specifically, we configure the perceptual data stream composed of different data with the same category for each view model to perform collaborative perceptual learning. The relatively independent perceptual data environment we construct may bring more supervision information to each view model than SSL, so we compare the model’s performance with the fully-supervised learning method in this experiment.

**Datasets** We configure multiple sets of experiments with different amounts of labeled data on the CIFAR-10/100 [42] and SVHN [43] datasets, which are widely used for SSL methods. Both CIFAR-10 and SVHN are 10-class datasets, where CIFAR-10 contains 50,000 training images and 10,000 test images with class balance, and SVHN contains 73,357 training images and 26,032 testing images with imbalanced classes. CIFAR-100 is a relatively complex 100-class dataset containing 50,000 training images and 10,000 testing images, with only 500 training images for each class. In each group of experimental configurations, we extract a small amount of class-balanced data from the training set to construct the labeled dataset and remove the labels of all training set data to form the unlabeled dataset. Note that the unlabeled datasets of SVHN are imbalanced. The test set of each dataset is used to evaluate the performance of different methods.

**Data Normalization** We employ different data-normalization techniques for models from different views. The main methods include normalizing the pixels of each image to conform to the standard normal distribution by calculating the channel-wise mean and standard deviation of the training set images, normalizing the image pixel values to be between 0 and 1, between −1 and 1, etc.

**Data Augmentation** We employ two different data-augmentation methods, weak augmentation $A_{w}\left(\cdot \right)$ and strong augmentation $A_{s}\left(\cdot \right)$. Weak augmentation methods will randomly flip and crop images to generate different views’ perceptual data. The strong augmentation method adopts Randaugment [44], which will be used for consistency regularization in SSL training on the collaborative discriminant results.

**Base Model** We use Wide-ResNet [45,46] as the base model, and, for simpler CIFAR-10 and SVHN tasks, we use WRN-28-2. For the CIFAR-100 classification task with more categories, we use the wider WRN-28-8. Models from different views will use different parameter-initialization methods [47] to further increase the discriminant difference.

**Optimizer Settings** We adopt the unified SGD optimizer for all models, with momentum β=0.9 and Nesterov, and use weight decay with the coefficient of 0.0005. We uniformly set models to the initial learning rate η=0.05 and use the cosine learning rate decay [48] strategies.

We set a uniform number of iterations $K=2^{18}$ for each experiment. For each experimental configuration with different amounts of labeled data, we use different random seeds to sample three sets of labeled data for model training to ensure the reliability of the experimental results. The total number of different view models is *V*. All hyperparameters used in the experiments are reported in Table 1.

### 4.2. Main Results

We report the performance comparison of our method with the multi-model SSL methods Mean Teacher [32], Dual Student [33], Deep CT [24], Tri-net [25], and the representative single-model SSL methods UPS [49], MixMatch [18], and FixMatch [21]. We configure 2-view and 3-view MACP experiments to compare the difference in the number of views; each experiment was run three times with a different labeled data split, and the mean and standard deviation of the final test error rates are reported.

As Table 2 shows, as the past multi-model SSL methods only focus on the prediction consistency between different models or have shortcomings in constructing differentiated multi-view data, the final performance is weaker than other methods. The proposed self-supervised model-initialization method can make models from different perspectives have more independent feature-discrimination criteria, thus increasing the reliability of the fused discriminant results. Some state-of-the-art single-model SSL methods strengthen the consistency-regularization constraint, adopt more post-processing algorithms for the predicted distribution of unlabeled data to obtain more reliable pseudo-labels, and achieve better performance. However, these methods cannot directly realize multi-view model interaction in the OEL environment. Under the same hyperparameter configuration, our proposed MACP method outperforms other algorithms in both two-view and three-view experiments. In the class-imbalanced SVHN experiment, MACP also achieved better performance than other methods, indicating that the multi-view discriminative information fusion performs a more reliable class judgment on unlabeled data. Moreover, in the CIFAR-10-1000-label and CIFAR-100-4000-label experiments with relatively few labeled data, MACP achieves 5.29% and 31.67% test error rates, respectively, which are significantly better than other methods. These results show the superiority of MACP in the face of perceptual environments where labeled data are lacking.

We further analyze the training efficiency and stability of MACP, see Figure 4. As Figure 4a shows, in the CIFAR-10-1000-labels experiment, the convergence efficiency and test accuracy of MACP are significantly higher than MixMatch and FixMatch, thus achieving a better final performance. Since the methods used for comparison are single-model SSL, to ensure the fairness of the comparison, we did not use the integrated prediction results of multi-view MACP for the performance evaluation but reported the independent evaluation results of each view model separately. The training curve of one view model in MACP is displayed normally in Figure 4, and the training curves of the other view models are represented by thin transparent curves. We also evaluate the performance of multi-view discriminative information fusion during MACP training. We find that, in the later training stage, MACP can obtain more than 90% unlabeled perception data in each batch for SSL training, and the pseudo-label accuracy of these data is higher than 97.5%, which shows that the DIF process of MACP generates more reliable pseudo-labels, thus achieving a more stable learning performance. If we follow the ensemble learning strategy and fuse the prediction results of different view models on the test set, the test accuracy will be further improved. However, to ensure a fair comparison, we still use the results of the single-view model to compare with existing methods. Furthermore, MACP’s performance under different numbers of views is evaluated. As can be seen from Figure 4b, the 3-view MACP can achieve higher convergence efficiency and test accuracy than the 2-view version during the whole training process. We find that, with the increase of views, due to the addition of more independent discriminative information, the reliability of the DIF of the models will be continuously improved, and the generalizability of models from different views will be jointly enhanced.

### 4.3. Ideal Multi-View Perception Experiments

The MACP method assumes that each view model continuously obtains perception data from different viewpoints. These view-specific perception data can be used for model training through the DIF process. The multi-view perception data in real scenes are naturally quite different. However, in the experiments above, we perform random data-augmentation methods on the same data to generate simulated multi-perspective data, which may still contain more related information, thus limiting the model’s performance.

More differentiated multi-view perception data will help each view model learn more independent feature-extraction patterns. To strengthen the difference between perception data from different views, we design a new sensing environment to test the online evolutive learning of multi-view agents under ideal conditions. Specifically, for each iteration, we assign a batch of different data with the same category to each view model. The joint discrimination results of each view model on these perceptual data will be used for SSL training.

In Table 3, we report the performance comparison of MACP and the fully-supervised learning method with various configurations under the ideal perceptual environment. In a more independent perception environment, since models from different views can provide more differentiated discriminant information, the reliability of the final DIF result will be significantly enhanced, providing more accurate classification supervision for each model. At the same time, through the information exchange between multi-view models, each model can obtain more generalized knowledge, which significantly reduces the model variance and enables MACP to achieve a better performance than fully-supervised learning. These results show that MACP has great application potential in practical multi-view sensing environments, which can solve the labeled data scarcity problem for edge models and enable each agent to adapt to the sensing-environment changes continuously.

In Figure 5a, we compare the training curves of MACP and the fully-supervised method under the CIFAR-10 databset. It can be seen that MACP has a faster convergence speed than fully-supervised learning, and the training fluctuation is smaller, which indicates that the multi-view models’ collaborative perception makes the training process more stable. As shown in Figure 5b, the stability advantage of MACP is more pronounced in the more complex CIFAR-100 experiments. In the early training stage of each view model, a small number of discriminative fusion results can be extracted from the easy-to-judge perceptual data. As the training progresses, the discriminative ability of each view model is gradually enhanced, and more valuable information can be obtained from more perceptual data. Such a step-by-step training process prevents the models from converging to local minima, resulting in better performance.

### 4.4. Ablation Study

Since the proposed MACP method consists of three modules, self-supervised model initialization (SMI), discriminative information fusion (DIF), and parameter constraints (PC), we will further analyze the impact of different module combinations on model performance.

We report the performance of 3-view MACP in CIFAR-10-4000-labels, SVHN-1000-labels, and CIFAR-100-10,000-label experiments under various module combinations. The DIF method is the key to the multi-view collaborative perception system. SMI and PC will provide differential regularization for each view model in the initial training stage and the subsequent training process, respectively, to ensure each view model’s relatively independent discriminative ability.

As shown in Table 4, when the three modules are not applied, it is equivalent to training each model separately without any information exchange, and the performance of each view model is poor at this time. When only DIF is used, due to the lack of independence constraints between models, the discriminative pattern of each model will gradually converge during the training process, resulting in performance bottlenecks in the later training stage. When using SMI combined with DIF, due to the lack of continuous independence constraints of models in the later training stage, it faces the risk of falling into the plateau as the model iterations, although it has high convergence performance in the early training stage. When DIF and PC are combined, although the initial independence constraint is lacking, each view model can gradually improve the difference in feature-discrimination patterns during the training process, thus achieving better performance. When the three modules are applied together, each view model can continuously exchange information in a relatively independent discriminative environment and achieve the optimal final performance.

In Section 3.4, our discriminative information-fusion method is implemented in a synchronous manner, and the DIF results of different view models on unlabeled perceptual data will be directly used for their respective training. Here we further explore the impact of the asynchronous DIF approach on model performance. In the asynchronous DIF setting, the parameter updates of each view model will be performed sequentially, that is, after the current model is updated according to the current DIF results, the model of the following view will be trained using the updated DIF results.

Through extensive experiments, we found that the asynchronous DIF approach is not conducive to optimizing MACP. Although asynchronous DIF enables a faster transfer of discriminative information between models, there is an increased risk of mis-discrimination. In the early training stage, when the discriminative ability of each perspective model is insufficient, there may be more misjudgments in the collaborative discrimination results. At this time, the alternate training of models will lead to the continuous accumulation of training errors, weakening the stability of DIF results. In asynchronous processing, the current model needs to wait for the update of other view models, which also affects the training efficiency. Moreover, the asynchronous DIF method will increase the possibility of the discriminative pattern convergence between models during the training process, affecting the discriminative independence of the models from different views. Overall, the asynchronous DIF approach will result in an up-to-5% performance degradation for each view model.

## 5. Conclusions

We propose MACP, which realizes online evolutive learning for efficient adaptation to the continuously changing sensing-data distribution through multi-view models’ independence constraints and collaborative discrimination. MACP consists of three main modules. Through the self-supervised model-initialization method, each view model learns different feature-extraction patterns. Through the discriminative information-fusion process, more reliable pseudo-labeled predictions are mined from multi-view unlabeled perceptual data for SSL training. Combined with the multi-model parameter constraint during the training, MACP achieves excellent performance over multiple representative multi-model and single-model SSL methods. In experiments on simulated ideal multi-view perception environment, MACP achieves performance that surpasses the fully-supervised learning methods, proving the practical application value of the proposed method.

With the increase of various edge intelligent sensing devices, the online evolutive learning method that improves the continuous adaptability of edge models to environmental changes through multi-agent collaboration will have significant developmental prospects. In future work, we will investigate more multi-agent interactive learning methods and discriminant independence-constraint methods to improve the adaptability and generalization of edge models to the perception environment. We will also explore the application of MACP in the real-world perception environment.

## Figures and Tables

**Figure 1 sensors-22-06893-f001:**
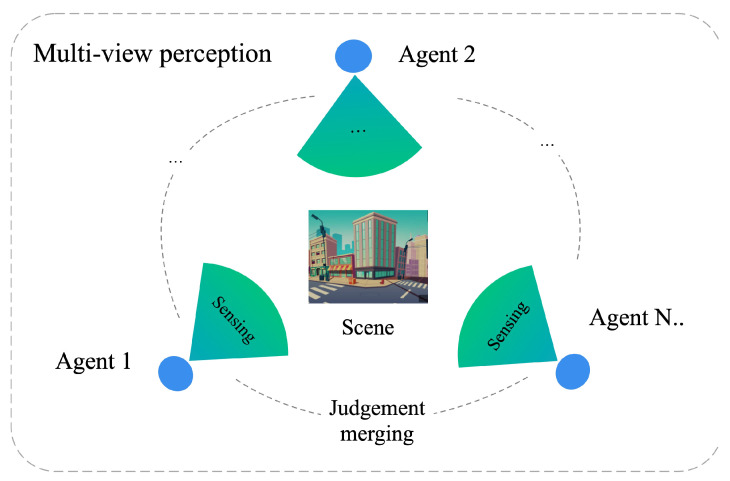
Diagram of multi-view agents collaborative perception and discrimination process.

**Figure 2 sensors-22-06893-f002:**
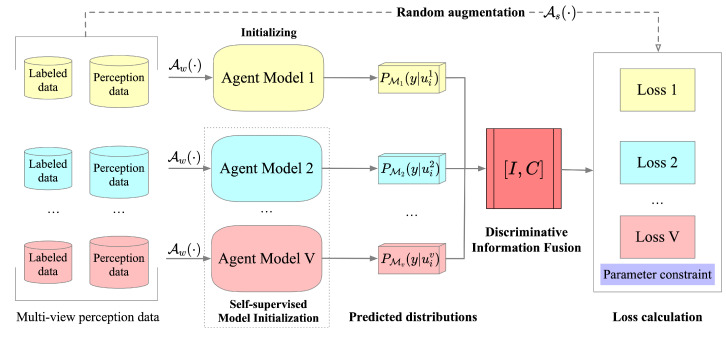
The overall framework of MACP.

**Figure 3 sensors-22-06893-f003:**
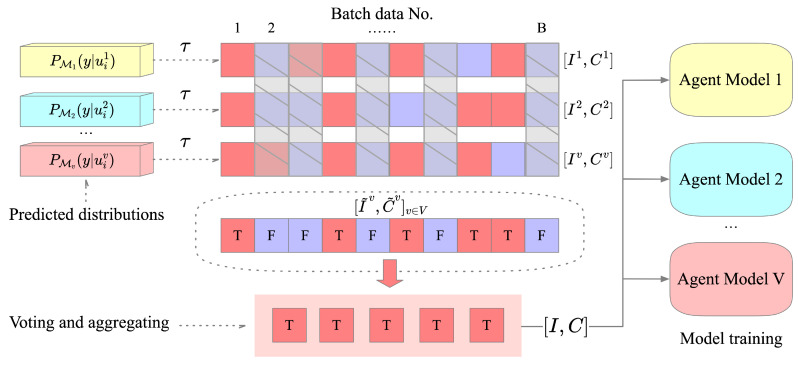
Multi-view agents discriminative information-fusion process.

**Figure 4 sensors-22-06893-f004:**
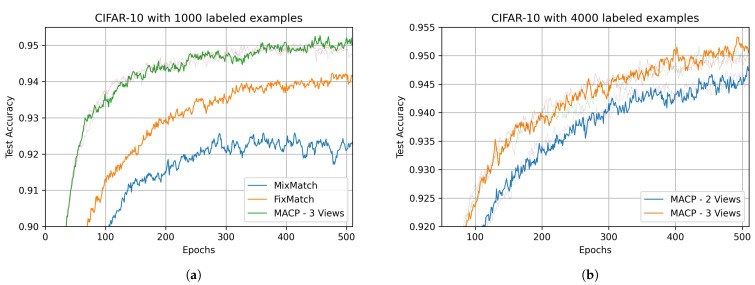
Performance comparison of MACP with different methods and different number of views during training. (**a**) Performance comparison for MixMatch, FixMatch and MACP on CIFAR-10-1000-label experiment; (**b**) Performance comparison for 2-view and 3-view MACP on CIFAR-10-4000-label experiment.

**Figure 5 sensors-22-06893-f005:**
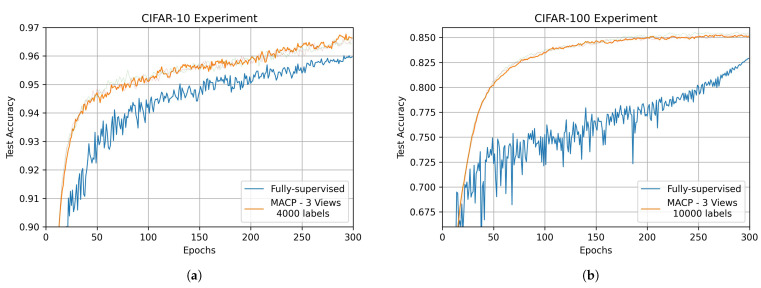
Performance comparison between MACP and fully-supervised method. (**a**) Performance comparison of 3-view MACP and fully-supervised method on CIFAR-10 experiment; (**b**) Performance comparison of 3-view MACP and fully-supervised method on CIFAR-100 experiment.

**Table 1 sensors-22-06893-t001:** List of hyperparameters for all datasets.

Dataset	CIFAR-10	SVHN	CIFAR-100
τ	0.95		
*V*	[2, 3]		
μ	4		
*B*	64		
*K*	$2^{18}$		
η	0.05		
β	0.9		
Weight decay	0.0005		

**Table 2 sensors-22-06893-t002:** Comparison of error rate (%) for CIFAR-10/100 and SVHN on three different labeled data folds, the comparison methods are tested under the same codebase.

	CIFAR-10			SVHN		CIFAR-100	
**Method**	**1000 Labels**	**2000 Labels**	**4000 Labels**	**250 Labels**	**1000 Labels**	**4000 Labels**	**10,000 Labels**
Mean Teacher	21.55 ± 1.48	15.73 ± 0.31	12.31 ± 0.28	4.35 ± 0.50	3.95 ± 0.19	-	-
Dual Student	14.17 ± 0.38	10.72 ± 0.19	8.89 ± 0.09	4.24 ± 0.10	-	-	33.08 ± 0.27
Deep CT	-	-	8.54 ± 0.12	-	3.38 ± 0.05	-	34.63 ± 0.14
Tri-net	-	-	8.30 ± 0.15	-	3.45 ± 0.10	-	-
UPS	8.18 ± 0.15	-	6.39 ± 0.02	-	-	40.77 ± 0.10	32.00 ± 0.49
MixMatch	7.72 ± 0.37	6.89 ± 0.39	5.21 ± 0.09	4.06 ± 0.18	3.49 ± 0.32	36.12 ± 0.62	29.12 ± 0.34
FixMatch	6.18 ± 0.56	5.92 ± 0.32	4.99 ± 0.11	3.83 ± 0.45	3.08 ± 0.63	33.78 ± 0.31	25.69 ± 0.61
MACP (2 views)	6.02 ± 0.39	5.69 ± 0.40	4.91 ± 0.08	3.57 ± 0.34	2.99 ± 0.26	33.52 ± 0.45	25.77 ± 0.83
MACP (3 views)	5.29 ± 0.37	5.12 ± 0.31	4.75 ± 0.20	3.32 ± 0.51	2.72 ± 0.15	31.67 ± 0.29	24.72 ± 0.11

**Table 3 sensors-22-06893-t003:** Comparison of test accuracy (%) between MACP and fully-supervised method in an ideal multi-view perception environment.

	CIFAR-10		SVHN		CIFAR-100
**Method**	**1000 Labels**	**4000 Labels**	**250 Labels**	**1000 Labels**	**10,000 Labels**
Fully-supervised	95.98		97.72		82.82
MACP (2 views)	96.23 ± 0.12	96.45 ± 0.07	97.82 ± 0.31	98.16 ± 0.51	83.06 ± 0.18
MACP (3 views)	96.41 ± 0.21	96.75 ± 0.03	98.21 ± 0.17	98.38 ± 0.34	85.39 ± 0.11

**Table 4 sensors-22-06893-t004:** Ablation study on MACP with different module combinations, test accuracy (%) of each setting on CIFAR-10/100 and SVHN are reported.

Module Combination			Dataset		
**SMI**	**DIF**	**PC**	**CIFAR-10**	**SVHN**	**CIFAR-100**
			92.10 ± 0.72	95.15 ± 0.34	72.17 ± 0.53
	✓		93.17 ± 0.19	95.98 ± 0.12	73.97 ± 0.43
✓	✓		94.23 ± 0.09	96.53 ± 0.31	74.92 ± 0.19
	✓	✓	94.93 ± 0.36	97.03 ± 0.22	74.62 ± 0.25
✓	✓	✓	95.25 ± 0.20	97.28 ± 0.15	75.28 ± 0.11

## Data Availability

The CIFAR-10/100 and SVHN datasets used for training and testing are available at https://www.cs.toronto.edu/~kriz/cifar.html and http://ufldl.stanford.edu/housenumbers/ (accessed on 24 July 2022).

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
