# Peer review of "Multi-Agent Multi-View Collaborative Perception Based on Semi-Supervised Online Evolutive Learning"

_sensors, 2022, doi:10.3390/s22186893_

Round 1
Reviewer 1 Report
This paper proposes an online, semi-supervised evolutionary learning method for collaborative perception of multi-viewing agents (MACP) learning method. The research project proves to be interesting and innovative.
However, the authors should improve the manuscript with respect to these weaknesses:
1) The organization of sections in the manuscript should be revised. A Material and Method section containing the methodology the dataset and the design of the experiments would help the reader.
2) The description of methodology in section 3.4 should be improved, it is convoluted when reading.
3) Other metrics should also be reported in the results, not just accuracy or error rate.
4) The discussion of the results should be thorough.
5) Tests on unbalanced datasets should be conducted.
Reviewer 2 Report
This manuscript proposed a multi-view agents collaborative perception (MACP) semi-supervised online evolutive learning method. Generally, the ideal that exploiting and fusing multiply different view data are interesting and meaningful. There are some issues to be addressed.
1. What does the meaning of "Online evolution learning". How to realize the online learning? In my opinion, there are lack of solid learning strategy and experimental scheme to make its "online".
2. It is stated that multiple models with different initialization parameter strategy could improve the learning performance. But as we know the deep neural network usually obtain good learning performance with fixed initialized parameters. Did the different parameter initializations really have large effect on the constructed model training? Why?
3. In the proposed method, multiple views from a single view data are constructed. Then multiply different models are adopted for learning. In this case, multiple views refer to different data argumentation/normalization strategy. Why not combine those together into one patch, which is usually used in image classification tasks.
4. The manuscript focus on the perception data from different multi-view edge sensing devices. But the experimental set just uses the CIFAR-10/100, SVHN datasets which are the single-view datasets. The validation of the constructed method is still not sufficient.
5. From table 2, it can be seen that the improvements of the proposed method are tiny in comparisons with existing FixMatch method. If some additional influence factors such as complexity and computational cost are considered, the superiority can be ignored.
6. It is strange why the model is called "Agent model". What are the differences between the constructed agent model and the traditional self-supervised learning model?
Reviewer 3 Report
The authors propose a multi-view agents collaborative perception (MACP) semi-supervised online evolutive learning method.
The quality of the paper is good, as well as the presentation of contents. Both methods and results achieved are clearly described, showing how much the presented work is good.
Author Response
We greatly appreciate your recognition and encouragement.
Round 2
Reviewer 1 Report
-
Reviewer 2 Report
All my issues have been addressed.